# Potential Distribution and Medicinal Uses of the Mexican Plant *Cuphea aequipetala* Cav. (Lythraceae)

Luis Rafael Garibay-Castro [1], Pedro Joaquín Gutiérrez-Yurrita [2], Alma Rosa López-Laredo [1], Jesús Hernández-Ruíz [3] and José Luis Trejo-Espino [1,*]

[1] Centro de Desarrollo de Productos Bióticos, Instituto Politécnico Nacional (CEPROBI-IPN), Carr. Yautepec-Jojutla km 6, Calle CEPROBI No.8, Col. San Isidro, Yautepec C.P. 62739, Mexico; lgaribayc1400@alumno.ipn.mx (L.R.G.-C.); arlopez@ipn.mx (A.R.L.-L.)

[2] Centro Interdisciplinario de Investigaciones y Estudios sobre Medio Ambiente y Desarrollo, Instituto Politécnico Nacional (CIIEMAD-IPN), Calle 30 de Junio de 1520 s/n, Barrio la Laguna Ticomán, Mexico City C.P. 07340, Mexico; pgutierrezy@ipn.mx

[3] División de Ciencias de la Vida, Universidad de Guanajuato, Carretera Irapuato-Silao km 9, Ex Hacienda El Copal, Centro, Irapuato C.P. 36000, Mexico; hernandez.jesus@ugto.mx

* Correspondence: jtrejo@ipn.mx

**Abstract:** Carrying out studies that lead us to obtain information on both the cultural and biological heritage of a locality, region, or country allows us to create appropriate strategies for the conservation of biocultural diversity. In this context, the objective of this study was to model the potential distribution of *Cuphea aequipetala* Cav. within the Mexican territory, to identify the main environmental variables that delimit its habitat, and to obtain information from traditional knowledge through the medicinal uses that the inhabitants of nearby communities give to the plant. Potential distribution modeling was performed with MaxEnt together with 19 bioclimatic variables of Worldclim plus three variables closely related to the habitat of the species. Data on its presence were obtained in situ within the Lagunas de Zempoala National Park and from iNaturalist. Information on the medicinal uses of the plant was obtained through semi-structured surveys with people who were in continuous contact with it. The potential range of *C. equipetala* Cav. was 3205.63 km$^2$, which represents 0.16% of the Mexican territory. The altitude, precipitation in the driest period, average temperature of the warmest quarter, and average temperature of the driest quarter were the variables that had the greatest effects on the potential distribution (49%), and these factors mainly delimited the suitability of the habitat. *C. aequipetala* Cav. is still used in traditional medicine, mainly for conditions related to cancer, shocks, and inflammation. Finally, it was found that the potential distribution coincided with the states of the country where its medicinal use was reported. This information is important, since it constitutes the basis for performing actions targeting the conservation of this species of medicinal relevance. For example, potential distribution areas can be integrated into habitat restoration and conservation plans to prevent anthropogenic activities, such as felling, that directly affect the habitat. This information can also be used as a reference to promote the preservation of medicinal uses among the young population of the distribution areas.

**Keywords:** biodiversity; cancer weed; habitat suitability; ecological niche; biocultural diversity

## 1. Introduction

The alarming loss of biodiversity that we are currently facing is caused by several factors. Among the main ones are habitat degradation and the overexploitation of species [1]. In this context, it is of the utmost importance to make urgent decisions aimed at avoiding the loss of biodiversity. Parallel to this process of biological loss is the loss of cultural heritage, and both processes are interdependent as they share common threats. The loss of biodiversity also represents the loss of important natural elements that are part of the ancestral traditions and practices that take place in the various cultures of the world. Similarly, if

the loss of a culture and/or language occurs, the consequence is the loss of valuable information on the use and conservation of biodiversity. Culture and biological diversity make up the biocultural diversity of countries [2]. For their conservation, it is necessary to carry out studies that allow us to obtain information on the habitats in which different biological species develop and the possible correlations between them and cultural diversity [3].

Vidal and Brusca [4] mentioned that Mexico ranks first in the world in terms of linguistic diversity, with 364 registered languages; however, 13 have already disappeared and 64 are at very high risk of disappearing. They also reported that Mexico ranks fourth in the world in terms of biological diversity; however, it currently has 1213 species of flora and fauna in danger of extinction. Traditional knowledge about the medicinal uses of various plant species is an important component of the culture of the different indigenous groups present in Mexico [5]. This cultural trait is of great importance because it is directly related to the conservation of biodiversity [6]. Gavin et al. [7] proposed that biocultural approaches to conservation can achieve effective conservation outcomes if they address the erosion of cultural and biological diversity. Therefore, we hypothesize that knowing both the traditional uses that members of a human community give to medicinal plants, as well as the spatial and climatic variables that define the distribution of them, is crucial for the conservation of biocultural diversity.

Potential distribution models of species can be based on the concept of environmental niches, defined as the environmental conditions and resources that are necessary so that the individuals of a population can keep it viable [8]. Under this premise, models are generated from data about the presence of a species, along with the meteorological and environmental data (which function as predictors) of a given place [9]. Given that biotic and abiotic factors are in constant interaction and delimit the distribution of species, these models are useful for determining the bioclimatic variables that limit the habitat of organisms, helping us to understand the adaptability and climatic preferences of species and generate strategies for biodiversity conservation [10]. They are also crucial to explain certain ecological events, indicate the degree of suitability in the habitat for the development of populations of a specific species or a community, and even to infer models of speciation [9,11].

There are different mathematical models that use statistical methods to make predictions about species distribution; some of them are multiple regression or multivariate models [12], while others use existing records of species' presence and environmental information to generate bioclimatic profiles, for example, BIOCLIM [13] or the genetic algorithm for rule-set prediction (GARP), which look for non-random relationships between the environmental characteristics of the localities and the species [14]. Maximum entropy (MaxEnt) is a habitat suitability model that uses the principle of maximum entropy to calculate the most probable geographic distribution for a species; therefore, it estimates the probability of occurrence of the species [15]. Compared to other probabilistic models, MaxEnt is one of the most used since it is more tolerant of sampling bias, small samples, irregular sampling, and data with few site deviations, and has a higher prediction accuracy [16].

Several authors have used the MaxEnt model to conduct conservation-oriented species distribution studies. Wu et al. [17] used it to conduct a study in the mountainous area of Beijing, China, where they simulated the geographical distribution of the area with abundant plant diversity in the understory and analyzed the contribution of habitat factors to the probability of existence of an area with abundant plant diversity in *Patlycladus orientalis* L plantations. Kumar et al. [18] used MaxEnt to predict potential suitable areas for the medicinal tree *Oroxylum indicum* L. under current and future climatic conditions under three different scenarios. In another study, Tena et al. [19] modeled the potential distribution of *Malpighia mexicana* A. Juss in two geographic areas of Mexico using MaxEnt and species records from databases, local herbariums, and records collected by the authors, as well as climate variables representing long-term average, variable, and extreme temperature and precipitation conditions.

*Cuphea aequipetala* Cav. is a Mexican medicinal plant with a wide distribution, mainly in the central and southern areas. Outside the country, it has been recorded up to Guatemala.

It is popularly known as cancer weed, blow weed, or weed that arises from the water. It has been used in traditional Mexican medicine since the 16th century [20] for the treatment of gastrointestinal problems, inflammation, dermatological conditions, and symptoms associated with skin cancer [21,22]. Various studies with extracts and compounds produced by this plant have demonstrated its antioxidant, cytotoxic, gastroprotective, antimicrobial, anti-Helicobacter pylori, antinociceptive, and anti-inflammatory biological activities [23–30]. These biological activities are attributed to the secondary metabolites it produces, including mannitol, sesquiterpene lactones, tannins, coumarins, alkaloids, flavonoids, and phenolic compounds [24,25,28,31]. *C. aequipetala* Cav. grows in semi-warm, semi-dry, and temperate climates at an altitude of approximately 3000 m above sea level, mainly in pine, oak, or mixed (pine–oak) forests; tropical deciduous forests; and scrub and grassland areas [24]. However, the significant environmental factors that allow us to determine its distribution are unknown. In addition, it is not known which of them are responsible for higher population densities in certain areas of Mexico [31]. Some of the main threats that this plant faces in terms of its conservation are the collection of wild plants to be sold or for medicinal use, as well as the degradation of its habitat by anthropic activities, such as felling, grazing, pollution, and the use of land for crops.

The aim of the present study was to model the potential distribution of *C. aequipetala* Cav. within the Mexican territory, identify the main environmental variables that delimit its habitat, and obtain information from traditional knowledge through the medicinal uses that the inhabitants of nearby communities give to the plant. This will allow us to obtain important information to generate strategies for the conservation of biocultural diversity.

## 2. Materials and Methods

### 2.1. Study Area

*Cuphea aequipetala* Cav. is distributed in Honduras, Guatemala, and Mexico [24]. In the case of Mexico, it has been reported that it can be found throughout the national territory [31]. Therefore, the geographical scale considered for the present study was the entire territory of the Mexican Republic, located within the following geographical coordinates: (West) −118.365119934082, (East) −86.7104034423828, (North) 32.7186546325684, and (South) 14.5320978164673. The Mexican territory is made up of five tectonic layers (North American, Pacific, Rivera, Cocos, and Caribbean), whose interaction has led to the formation of mountain ranges through volcanism (Sierra Madre Occidental and Trans-Mexican Volcanic Belt) or by folding (Sierra Madre Oriental and Sierra Madre del Sur) [32]. The climatic diversity of Mexico ranges from dry climates in the north to humid and sub-humid climates in the south. On the Pacific slope, dry and sub-humid climates predominate, and it is humid with rain all year on the slope of the Gulf of Mexico. The coasts and depressions have tropical climates, while there are temperate and cold climates in the mountains, which can exceed 4000 m above sea level [32]. Such climatic conditions are due to the shape of the Mexican territory (wide in the north and narrow in the south), the mountainous systems, the action of the trade winds, and the seasonal oscillation of the subtropical high-pressure belt [33–35]. Biogeographically, Mexico is a transition zone between the Nearctic region that comprises the northern part and the highlands of the country, together with the mountains that surround them and the Neotropical region that is constituted by the southern part of the country and the Yucatan Peninsula [36]. This has led to an exchange of biota between the north and south of the continent, and, therefore, the diversification of flora and fauna in Mexico [36].

### 2.2. Model of Potential Distribution and Environmental Variables

MaxEnt 3.4.1 software [37] was used to model the potential distribution of *C. aequipetala* Cav. This program models the geographic distribution of species with records of presence and determines the distribution through maximum entropy probability (close to uniform) delimited by a set of covariates known as climate, topography, and soils and vegetation [38]. Data on the known presence of the species (geographical coordinates) were considered for

a total of 215 plants, of which 90 were obtained in situ (with a field GPS navigator; brand Garmin, model etrex 22, and <3 m precision) in three different areas within the Lagunas de Zempoala National Park (Figure 1), and 125 were obtained in different regions of Mexico from records reported in the iNaturalist database (the criteria used validate the records in iNaturalist were: data with a maximum of 10 years of age; classified as a research-grade observation; and reported by a curator) [39]. The variables used for the present study were: 22 environmental variables and 19 bioclimatic variables with a spatial resolution of 30 s (~1 km$^2$), which were obtained from the WorldClim database v. 2.0 (Table 1) [40]. In addition, three variables closely related to the habitat of the species [41] were considered, namely altitude, soil moisture regimes, and soil and vegetation cover, which were obtained from the CGIAR-CSI [42] and CONABIO [43,44] databases, respectively. These 19 initial variables were selected because they are the ones that are usually used to develop prediction models [38]. The three additional variables were chosen because they are variables that determine the physical space that the species inhabits [41]. Particularly, the humidity regime delimits the dominant plant formation and establishes the abundance of populations.

**Table 1.** Environmental variables used for modeling the potential distribution of *Cuphea aequipetala* Cav. and its percentage of contribution according to the Jackknife analysis of the importance of the variables.

| Code | Description of the Variable | Percentage of Contribution |
|:---:|:---:|:---:|
| Bio20 | Altitude (m) | 25 |
| Bio14 | Precipitation in the driest period (mm) | 13.2 |
| Bio9 | Average temperature in the driest quarter (°C) | 10.8 |
| Bio10 | Average temperature in the warmest quarter (°C) | 9 |
| Bio4 | Temperature seasonality (CV) | 6.6 |
| Bio21 | Soil moisture regime | 5.4 |
| Bio19 | Precipitation in the coldest quarter (mm) | 5.3 |
| Bio7 | Annual temperature variation (°C) | 5 |
| Bio15 | Precipitation seasonality (CV) | 4.4 |
| Bio2 | Diurnal temperature variation (°C) | 3.6 |
| Bio3 | Isothermality (dimensionless) | 3.5 |
| Bio12 | Annual precipitation (mm) | 2.6 |
| Bio11 | Average temperature in the coldest quarter (°C) | 1.8 |
| Bio6 | Average minimum temperature in the coldest period (°C) | 1.2 |
| Bio22 | Land cover (23 types) | 0.9 |
| Bio1 | Average annual temperature (°C) | 0.7 |
| Bio17 | Precipitation in the driest quarter (mm) | 0.5 |
| Bio16 | Precipitation in the rainiest quarter (mm) | 0.2 |
| Bio5 | Average maximum temperature in the warmest period (°C) | 0.2 |
| Bio8 | Average temperature in the rainiest quarter (°C) | 0.1 |
| Bio18 | Precipitation in the warmest quarter (mm) | 0.1 |
| Bio13 | Precipitation in the rainiest period (mm) | 0 |

The results of the Jackknife analysis are shown in Figure S1.

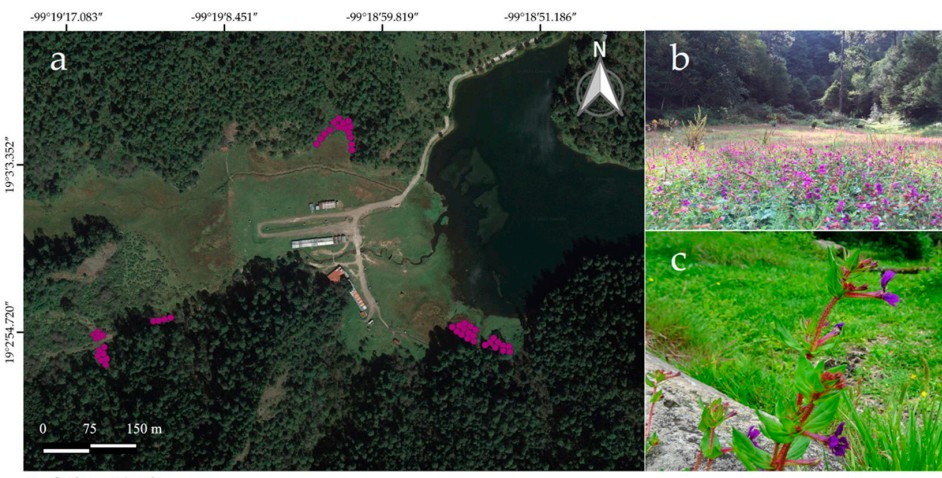

**Figure 1.** Wild populations of *Cuphea aequipetala* Cav. growing in the Lagunas de Zempoala National Park. (**a**) Map with the location of the populations in the recreation area; (**b**) wild population; (**c**) wild plant.

Due to the fact that MaxEnt determines the contribution of each variable to predict the potential distribution, we started from an initial model with the proposed variables (Table 1). Subsequently, to exclude highly correlated variables and avoid possible influences on the following analyses, a test of the normality and homoscedasticity of the data was performed before subsequently conducting an analysis of variance (in which the bioclimatic variables were considered as sources of variation) using the SAS 6.0.8 statistical package with a completely randomized design and applying Pearson's correlation coefficient [45]. This model provides results on the presence probability, with an interval from 0 to 1. Values close to the lower limit indicate that the data do not fit and that the prediction of the potential distribution is low or null, whereas a value close to 1.0 means that the potential distribution corresponds to an appropriate model with a high probability of the presence of the species [10,46].

The potential distribution of *C. aequipetala* Cav. (heat map) was transformed to GRD format to be georeferenced and processed with DIVA-GIS software [47]. Subsequently, the image obtained was taken as a reference to build a layer of polygons with the potential distribution of the highest probability of presence and describe its attributes through contrast analysis by the superimposition of three different thematic layers using digital cartographies in QGIS 3.6 Noosa [48]. The three digital maps used were digital mapping of vegetation, scale 1:4,000,000 [49]; digital mapping of climates (modified Köppen classification), scale 1:10,000,000 [50]; and digital mapping of soil moisture regimes, scale 1:4,000,000 [43], which were obtained from the Geoportal of the National System of Information on Biodiversity (SNIB) [51].

### 2.3. Medicinal Uses of Cuphea aequipetala Cav.

The present study was conducted using a semi-structured survey and partially following the strategy proposed by De Beer and Van Wyk [52]. In 2008, our working group located a wild population of *C. aequipetala* Cav. (Figure 1) within the Lagunas de Zempoala National Park (LZNP) [26], which has been monitored annually. For identification, plants were collected from wild populations located in the LZNP at 2860 m above sea level (latitude 19°02′ N, longitude 99°19′ W). The plants were positively identified as *Cuphea aequipetala* Cav. in the herbarium of the Universidad Autonoma del Estado de Morelos. The voucher number is 13,238. With this background and in order to ensure that most of the individuals to whom the survey would be given were familiar with the plant, we decided to select those who met two criteria: (i) being a resident of one of the localities of the State of Mexico or the State of Morelos bordering the LZNP; and (ii) individuals providing services

within the LZNP (handicraft sellers, food sellers, forest rangers, guides, or individuals who offered horseback riding) whose daily activities allowed them to be in contact with the biodiversity of the LZNP, particularly *C. aequipetala*.

The surveyed population was made up of 57 individuals (Table 2). The surveys were complemented with photographs of *C. aequipetala* Cav. that were shown to the individuals surveyed prior to asking the following questions: Do you know this plant? What is its name? Does it have any use or do you know if it has any use? How is it used? Where have you seen it and at what time of year? From whom or where did you obtain knowledge about it? Additionally, do you think it is important to conserve this plant and the knowledge about it? The data of each individual surveyed were recorded, i.e., their age, sex, cultural identity/language, place of residence, and level of schooling.

**Table 2.** Demographic, social, and cultural features of the people surveyed.

| | Features | Frecuency | % |
|---|---|---|---|
| Gender | Female | 32 | 56.14 |
| | Male | 25 | 43.86 |
| Age group | Children/teenagers (6–14 years old) | 3 | 5.26 |
| | Young (15–24 years old) | 14 | 24.56 |
| | Young adults (25–44 years old) | 21 | 36.84 |
| | Mature adults (45–59 years old) | 14 | 24.56 |
| | Elders (over 60 years old) | 5 | 8.77 |
| Education level | Basic | 10 | 17.54 |
| | Medium | 20 | 35.09 |
| | High–medium | 16 | 28.07 |
| | Technic | 2 | 3.51 |
| | University | 7 | 12.28 |
| | None | 2 | 3.51 |
| Cultural identity | Otomi | 2 | 3.51 |
| | Tlahuica | 15 | 26.32 |
| | NI [1] | 40 | 70.18 |
| Origin place [2] | Huitzilac, Huitzilac, Mor. | 15 | 26.32 |
| | Santa Lucía, Ocuilan, Edo. Mex. [3] | 12 | 21.05 |
| | San Jerónimo Acazulco, Ocoyoacac, Edo. Mex. [3] | 1 | 1.75 |
| | San Juan Atzingo, Ocuilan, Edo. Mex. [3] | 5 | 8.77 |
| | San Nicolás, Coatepec, Edo. Mex. [3] | 1 | 1.75 |
| | Santa Martha, Ocuilan, Edo. Mex. [3] | 14 | 24.56 |
| | Ocuilan, Ocuilan, Edo. Mex. [3] | 8 | 14.04 |
| | Toluca, Edo. Mex. [3] | 1 | 1.75 |

[1] NI: Not Identified; [2] The geographical position of the places of origin with respect the Lagunas de Zempoala National Park is in Figure S2; [3] Edo. Mex: Estado de Mexico.

## 3. Results

### 3.1. Potential Distribution of Cuphea aequipetala Cav.

The potential distribution was obtained through mathematical modeling using MaxEnt 3.4.1 software. Figure 2 illustrates the territory with the highest probability of presence (0.74–1.0) for *C. aequipetala* Cav. The areas in red show the ideal bioclimatic conditions for the development of this species. The area covers approximately 3205.63 km$^2$, which represents 0.16% of the Mexican territory.

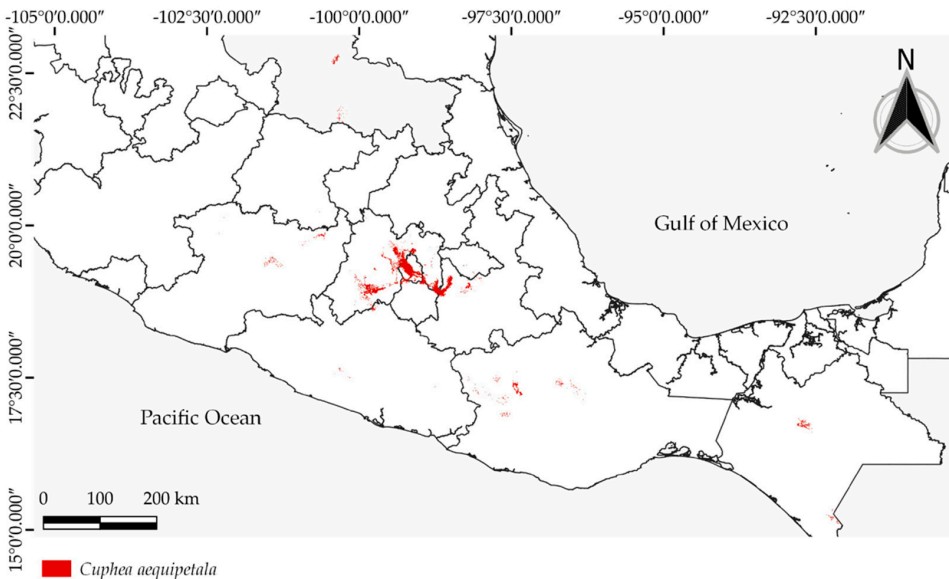

**Figure 2.** Potential distribution of *Cuphea aequipetala* Cav. in Mexico using a maximum entropy model. The red color indicates the polygons with the highest probability of presence (0.74–1.0).

The analysis of the potential distribution of *C. aequipetala* Cav. indicated a high predictive performance value (AUC) of 0.98 (Figure 3a). The response curves of the variables that contributed most to the suitability of the habitat for *C. aequipetala* Cav. are illustrated in Figure 3b–d. With respect to altitude (Figure 3b), it was observed that the maximum value of AUC (0.81) corresponded to an altitude of 3100 m above sea level. Regarding the precipitation in the driest month (Figure 3c), the greatest probability of presence (AUC of 0.62) corresponded to precipitation of 9 mm. On the other hand, the highest AUC value of the mean temperature in the driest quarter (Figure 3d) was 0.69, which corresponds to a temperature of 9.5 °C.

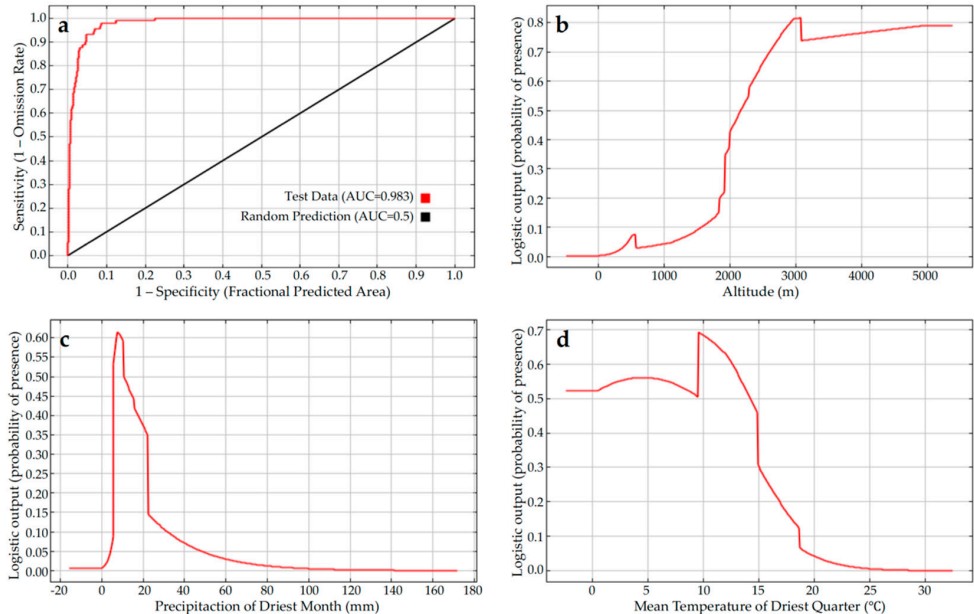

**Figure 3.** Curves showing the accuracy of the model. Curve of AUC for (**a**) environmental variables with the greatest contribution in the modeling of the potential distribution of *Cuphea aequipetala* Cav.; (**b**) altitude; (**c**) precipitation of the driest period; and (**d**) average temperature of the driest quarter.

The results of the jackknife analysis that determined the contribution of the variables (Table 1) indicated that, of the 22 used, the altitude (Bio20), the precipitation in the driest month (Bio14), mean temperature in the driest quarter (Bio9), and average temperature in the warmest quarter (Bio10) were the ones that had the greatest effect on the potential distribution (58% of the whole). Regarding the importance of the permutation, altitude (Bio20) was the variable that decreased the regularized training gain the most, which suggests that it had the greatest amount of information that was not present in the other variables. On the other hand, the variable with the greatest gain was the average temperature of the warmest quarter (Bio10), which suggests that it contained the most useful information.

The overlapping of layers between the potential distribution and the maps, both of vegetation (Figure 4) and the soil moisture regimes (Figure 5), indicated that *C. aequipetala* Cav. was mainly associated with coniferous and oak forests, as well as areas with ustic soils that retain moisture for 180 to 270 days a year. On the other hand, in contrast to the climate map for the Mexican Republic (Figure 6), it was observed that *C. aequipetala* Cav. was distributed mainly in areas where climate type C(w2) predominated.

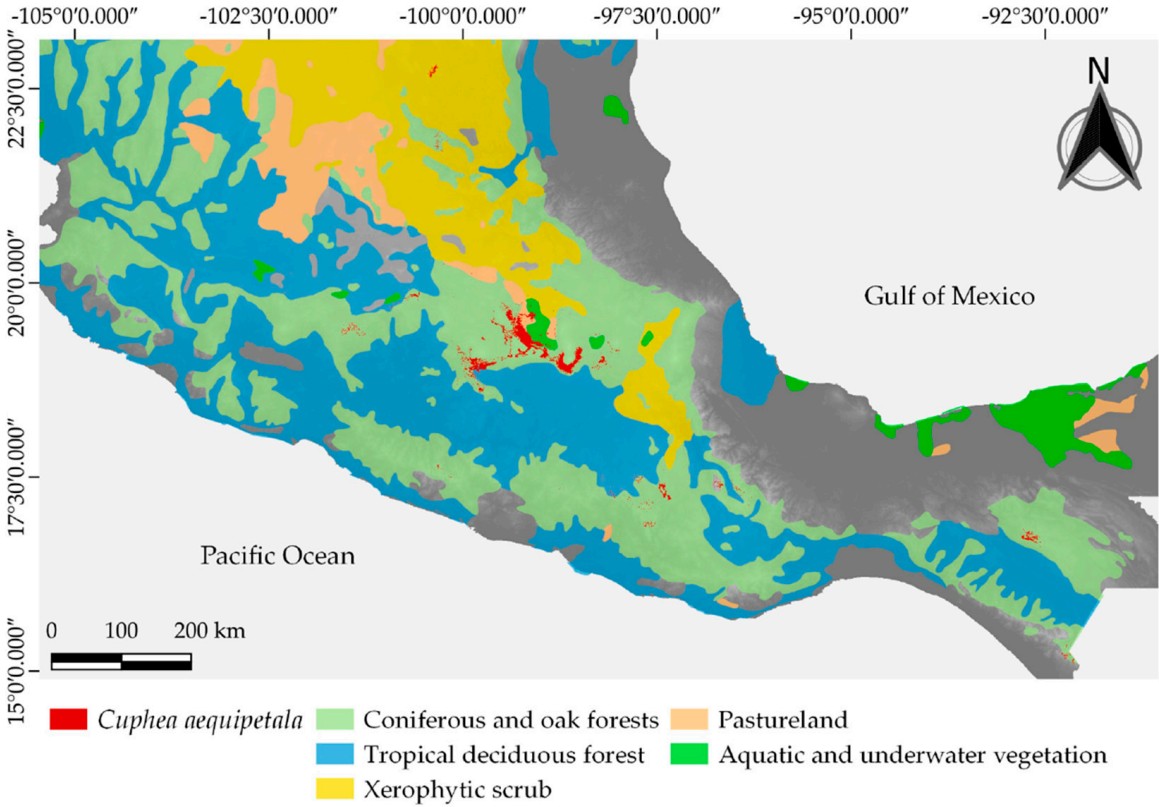

**Figure 4.** Contrast analysis created by overlapping the potential distribution of *Cuphea aequipetala* Cav. with a map of vegetation types in Mexico.

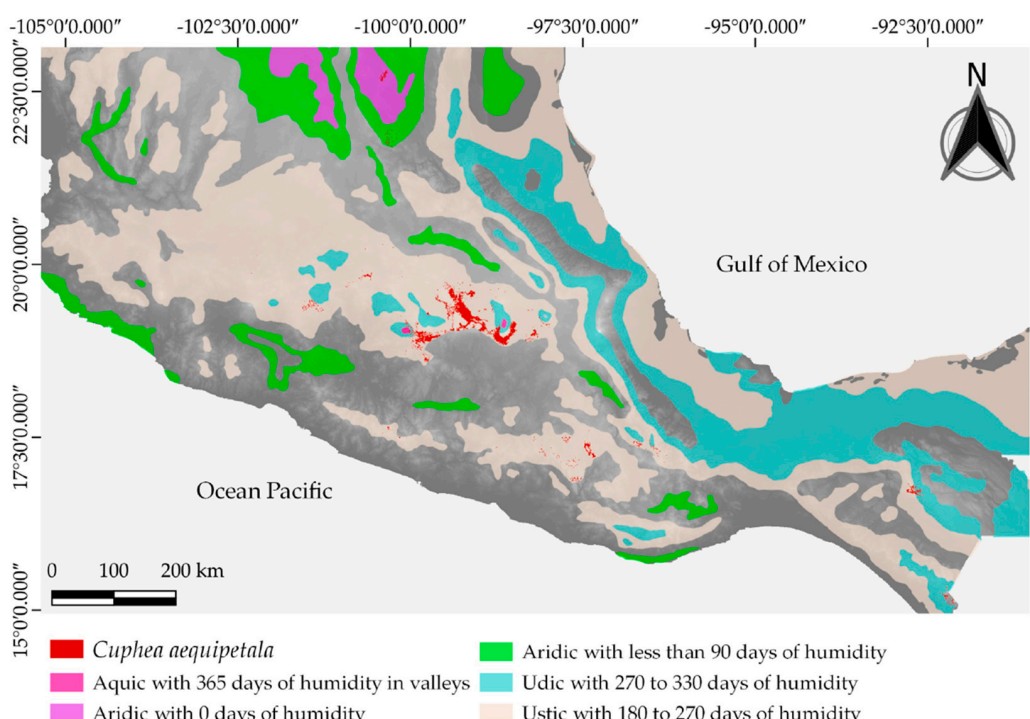

**Figure 5.** Contrast analysis created by overlapping the potential distribution of *Cuphea aequipetala* Cav. with a map of moisture regimes in Mexico.

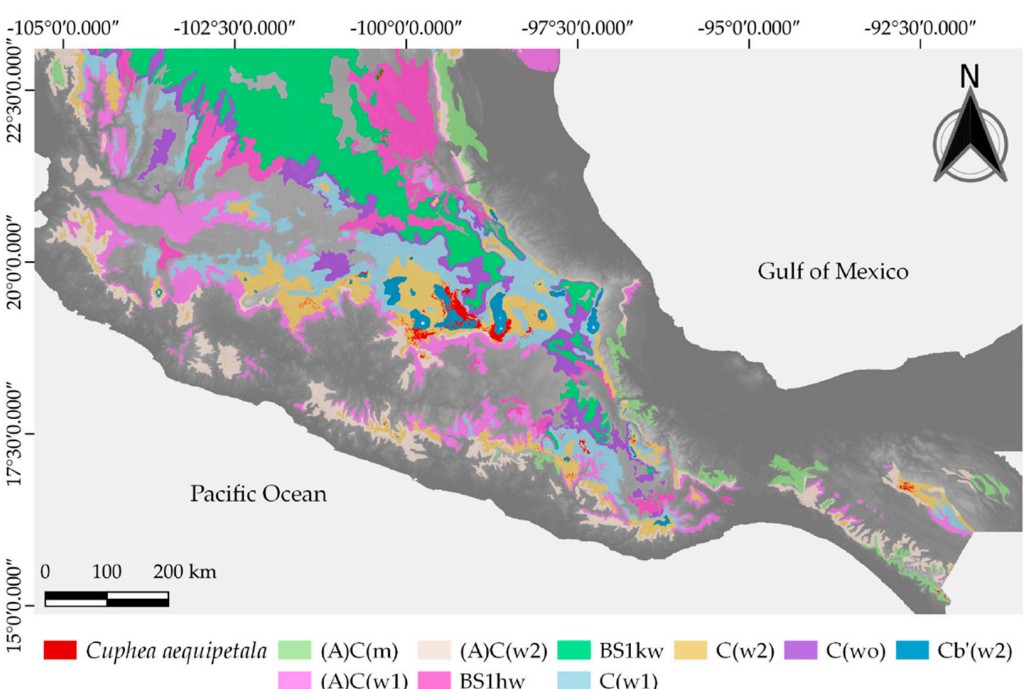

**Figure 6.** Contrast analysis by overlapping the potential distribution of *Cuphea aequipetala* Cav. with a climate map of Mexico. (A)C(m): semiwarm humid; (A)C(w2): semiwarm subhumid; BS1kw: temperate semiarid; C(w2): temperate subhumid with Lang's index greater than 55; C(wo): temperate subhumid with Lang's index less than 43.2; Cb′(w2): semicold subhumid; (A)C(w1): semiwarm subhumid; BS1hw: semiarid semiwarm; and C(w1): temperate subhumid, with Lang's index between 43.2 and 55 (Table S1).

*3.2. Ethnobotanical Knowledge about Cuphea aequipetala Cav.*

The results were organized by dividing the individuals surveyed by age group. The youngest were under 14 years old, while the oldest were over 65 years old. The majority (86%) answered that they knew or had seen the plant. Of these, 47% answered that they knew one or more common name used for the plant, and all those who knew of it and knew some common names confirmed that they were aware that it was used for medicinal purposes (42% of all respondents). The common names of *C. aequipetala* Cav. mentioned by the respondents were: '*hierba del cáncer* (cancer weed)' (mentioned 13 times); '*campanita* (bell)' (mentioned 3 times); '*hierba mora* (black nightshade)' (mentioned 2 times); '*cancerina*' (mentioned 2 times); '*lapida*' (mentioned 2 times); '*hierba de la fuerza* (strength weed)' (mentioned 1 time); '*flor de cáncer* (cancer flower)' (mentioned 1 time); and '*violeta* (violet)' (mentioned 1 time).

The uses, forms of use, and conditions treated with *C. aequipetala* Cav. are presented in Table 3. In general, the plant was used mainly for medicinal purposes, and only one person mentioned that it was used to obtain pigment "for children to paint". The forms of use were cooking the stems and leaves of the plant, or in the form of an infusion, and the majority mentioned that it was used to cure or prevent cancer and to reduce inflammation and pain. They also mentioned that they used it to treat gastrointestinal conditions or to improve healing.

**Table 3.** Medicinal uses of *Cuphea aequipetala* Cav., parts of the plant used, and forms of preparation mentioned by the people surveyed.

| Plant Part Used | Preparation Method | Medicinal Use | Frecuency of Citation |
|---|---|---|---|
| Aerial parts (flowers, leaves, and stems) | Decoction | Wound washing | 7 |
| | | Pospartum baths | 1 |
| | | To promote wound healing | 4 |
| | | Bumps and bruises | 1 |
| | Infusion | Headache, tooth, bone, or throat pain | 6 |
| | | Dizziness | 2 |
| | | Tumors | 2 |
| | | Stomach upsets and disorders | 2 |
| | | Sprain | 1 |
| | | Stress | 1 |
| | | Cough | 1 |

All the individuals who identified *C. aequipetala* Cav. through the photographs shown agreed that the plant could be found inside the park and in different areas of their localities where there was much humidity during the rainy season in the summer (specifically between June and October). Regarding from whom or where they obtained the knowledge that they provided, those who stated that they knew about the plant confirmed that they had acquired that knowledge orally through their relatives, generally from their parents or grandparents.

Regarding cultural identity, 30% stated that they belonged to an indigenous group; 15 individuals considered themselves Tlahuicas and 2 considered themselves Otomi. The respondents also agreed that conservation was important, both in terms of the places where the plant grew and the ethnobotanical knowledge about it (i.e., the transfer of medicinal knowledge about the plant for the treatment of ailments and as an alternative method to conventional medicine).

## 4. Discussion

The results of the modeling with MaxEnt were excellent, as indicated by the area under the curve (AUC) value of 0.98 (Figure 3a), which meant that the potential distribution found for *C. aequipetala* Cav. in the present study was statistically significant. This result is comparable to that reported by Dhyani et al. [53], who used MaxEnt to predict the potential distribution of the threatened medicinal plant *Lilium polyphyllum*. The accuracy of the modeling they found was considered very high, since the AUC value obtained was 0.98. Furthermore, Duno et al. [54] modeled the potential distribution of the genus Mappia in Mesoamerica and the Greater Antilles with MaxEnt, reporting an AUC value of >0.90.

The locations of the polygons that made up the potential distribution of *C. aequipetala* Cav. obtained in the present study (Figure 2) were mainly a strip found in the center of the country, in addition to some specific areas in the states of Chiapas, Oaxaca, San Luis Potosi, and Michoacán. These areas coincide with those corresponding to the Mexican mountainous massifs [55], which means that the populations were associated with places located 1000 m or more above sea level. This fact was corroborated by contrasting the potential distribution with the hypsometry map [56] for the Mexican Republic, in which it could be observed that the potential distribution zones for this plant were found at altitudes between 1500 and 3500 m above sea level.

Regarding the environmental variables, the results indicated that, of the 22 variables used, the altitude, precipitation in the driest month, and average temperature in the driest quarter were the ones that had the greatest effect on the potential distribution. These variables were the ones that mainly delimited the suitability of the habitat of *C. aequipetala* Cav. in Mexico. When contrasting the potential distribution area of *C. aequipetala* Cav. with the map provided by the Global Biodiversity Information Facility [57] based on the sighting data (taken since 1891) of *C. aequipetala* Cav., we found that the area was greater in the latter because it is made up of a historical accumulation of sightings, which model an area that, over time, has been disturbed or fragmented into different zones. Meanwhile, the potential distribution models a certain area based on the suitability of the bioclimatic variables using recent data (no more than 10 years old).

The altitude gradient found in the present study featuring higher probability of presence with a value of 0.8 (Figure 3b) (>3500 m above sea level) is broader than that reported by other authors [31,58], who stated that *C. aequipetala* Cav. was distributed mainly in areas with altitudes greater than 2000 m above sea level. It is also in line with the work of Waizel-Bucay et al. [24], who stated that *C. aequipetala* Cav. grew mostly at altitudes greater than 1000 m above sea level, but also that the areas with the greatest presence were found at altitudes around 3200 m above sea level.

On the other hand, the variables of precipitation in the driest month and average temperature in the driest quarter (Figure 3c–d) indicated that the highest probability of presence (0.62 and 0.7, respectively) occurred at intervals of precipitation from 9 to 20 mm, with temperatures ranging from 8 to 9 °C. The previous conditions refer to humid temperate climates with summer rains and sub-humid cold climates [33], which coincide with the results of the contrast analysis between the potential distribution and the climate layer for the Mexican Republic (Figure 5), indicating that *C. aequipetala* Cav. was distributed mainly in areas where the climate type C(w2) predominated, i.e., temperate, sub-humid, average annual temperature between 12 and 18 °C, temperature in the coldest month between −3 and 18 °C, temperature in the hottest month below 22 °C, precipitation in the driest month below than 40 mm, summer rains, Lang's index greater than 55, and percentage of winter rain from 5 to 10.2% of the annual total [33].

The overlapping of layers between the potential distribution and the maps, both of vegetation (Figure 4) and soil moisture regimes (Figure 5), indicated that *C. aequipetala* Cav. was mainly associated with coniferous and oak forests, as well as areas with ustic soils that retain moisture for 180 to 270 days a year. This finding is in line with what was reported by Graham [57] and Waizel-Bucay et al. [24], who observed that the plant grew in pine, oak, or mixed (pine–oak) forests, and with what was written by Francisco-Hernández [20], who



described *C. aequipetala* Cav. as developing in rural areas of temperate regions, such as that of 'Tetzcoco', in wet or watery lands, from where its name "weed that springs from the water" derives.

According to the Biblioteca Digital de la Medicina Tradicional Mexicana (BMTM) [59], the reported areas of use of *C. aequipetala* Cav. are in the states of Morelos, Puebla, Estado de Mexico, Mexico City, Michoacán, Hidalgo, Guanajuato, Chiapas, and Veracruz. However, the area in which *C. aequipetala* Cav. is potentially found is larger (because it is also found in San Luis Potosi, Oaxaca, Guerrero, and Tlaxcala, but not in Veracruz) (Figure S3). This is related to the incidence of the plant in these areas; that is, a greater incidence of a plant in a place could increase the probability that it has some type of use. On the other hand, areas without reports of use may be related to a low incidence of the plant or the fact that these areas have not been ethnobotanically evaluated.

The use of *C. aequipetala* Cav. as a medicinal plant in Mexico was described for the first time in the 16th century by Francisco Hernández [20]. The assessment of the information obtained from the surveys indicated that the ethnobotanical knowledge about the medicinal use of *C. aequipetala* Cav. was preserved among the inhabitants of the region near the Lagunas de Zempoala National Park, and that it has been transmitted orally through the generations (by their parents and grandparents). Despite this fact, it is worrying that the population aged over 54 years, and mainly individuals in the age group of over 60 years, were the ones who retained this knowledge, in addition to the fact that most of these individuals confirmed that they belonged to an indigenous group. It is thus noticeable that this knowledge, although conserved, was retained by a group with well-identified characteristics. This trend in the management of knowledge about the use of medicinal plants in Mexico is not unique to *C. aequipetala* Cav. A study conducted in the communities of the upper Mixteca region in Oaxaca, Mexico [6], described this same trend, i.e., knowledge about medicinal plants and their use was preserved in certain groups with well-defined sociocultural traits, i.e., older populations, female sex, from indigenous communities, with a low level of schooling, and without being migrants.

Of the eight common names with which *C. aequipetala* Cav. Was identified, five of them (campanita (bell), hierba de la fuerza (strength weed), lapida, cancerina, and *Hierba mora* (black nightshade)) were not reported in the bibliographical sources or revised databases [24,31,59–61]. The reason why these names are little known may be because they are local names and are used only by individuals who live in the communities surrounding the LZNP. In general, the common names by which *C. aequipetala* Cav. is known are closely related to the plant due to the shape and color of its flowers, as well as its medicinal uses.

Of the different medicinal uses of *C. aequipetala* Cav. mentioned by the individuals interviewed (Table 2), its use for the treatment of dizziness, 'jerks' (colloquial term used in Mexico to refer to stomach discomfort), stress, and coughs, in addition to non-medicinal use (for painting), have not been reported before; however, these are terms from which we can draw inferences, i.e., they refer to treating conditions that have been reported; for example, jerks could be related to treating stomach conditions. Its use and preparation coincide with what has been reported in other studies [24,59–61]. These traditional uses of *C. aequipetala* Cav. have been experimentally tested using plant extracts to assess its antinociceptive and anti-inflammatory effects, antioxidant and antimicrobial activity, its effect on cancer cell lines, and its anti-*Helicobacter pylori* effect [27,29,30,62,63].

## 5. Conclusions

The present study allowed us to obtain information about the factors that affect the suitability of a habitat for *C. aequipetala* Cav. and its medicinal use in Mexico. The altitude, precipitation levels in the months with lower water availability, and average temperature in the driest quarter were the environmental variables that contributed most to the potential distribution model of *C. aequipetala* Cav. These factors mainly delimited the suitability of habitats, which corresponds to coniferous and oak forests, with temperate and cold sub-humid climates and ustic-type soils. Knowledge about its use in traditional medicine is still

applicable; however, it is focused within a group of individuals with specific characteristics. Therefore, it is necessary to register and disseminate information on its use in order to preserve this valuable knowledge.

This information is important, since it constitutes the basis for performing actions targeting the conservation of this species of medicinal relevance; for example, potential distribution areas can be integrated into habitat restoration and conservation plans, and anthropogenic activities that directly affect its habitat, such as felling, can be avoided. This information can also be used as a reference in LZNP management programs and to promote the preservation of medicinal uses among the young population of the distribution areas. In addition, it can be used to conduct studies related, for example, to the effect of future climate change on biodiversity in this type of habitat.

**Supplementary Materials:** The following supporting information can be downloaded at https: //www.mdpi.com/article/10.3390/d14050403/s1, Figure S1: Jackknife regularized test for *Cuphea aequipetala* Cav.; Figure S2: Geographical position of the places of origin with respect to the Lagunas de Zempoala National Park; Table S1: Characteristics of the climates in which *Cuphea aequipetala* Cav. is potentially distributed in Mexico; Figure S3: Potential distribution of *Cuphea aequipetala* Cav. based on the areas of medicinal use proposed by BDMTM.

**Author Contributions:** Conceptualization, L.R.G.-C., J.L.T.-E., J.H.-R. and P.J.G.-Y.; data collection, L.R.G.-C.; observation and collection in the field, L.R.G.-C., A.R.L.-L. and J.L.T.-E.; mathematical modeling, L.R.G.-C. and J.H.-R.; design and application of the survey, L.R.G.-C.; writing—original draft, L.R.G.-C. and J.L.T.-E.; writing—review and editing L.R.G.-C., J.L.T.-E., J.H.-R., A.R.L.-L. and P.J.G.-Y. All authors have read and agreed to the published version of the manuscript.

**Funding:** This research was financed by the Secretary of Research and Postgraduate Studies of the IPN through the project SIP 20220129.; L.R.G.C. received support from the IPN Research Training Institutional Stimulus Scholarship (BEIFI) and a scholarship for doctoral studies granted by the National Council for Science and Technology (Conacyt).

**Institutional Review Board Statement:** Not applicable.

**Informed Consent Statement:** Not applicable.

**Data Availability Statement:** Not applicable.

**Conflicts of Interest:** The authors declare no conflict of interest.

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
