# Peer review of "Potential Distribution and Medicinal Uses of the Mexican Plant Cuphea aequipetala Cav. (Lythraceae)"

_diversity, doi:10.3390/d14050403_

Round 1

Reviewer 1 Report

This article presents Potential Distribution and Medicinal Uses of the Mexican Plant Cuphea aequipetala Cav. (Lythraceae). This study will help to understand the medicinal uses and distribution of medicinal plants in a specific region. I suggest authors review the whole manuscript carefully and correct all the mistakes. Authors should improve the grammar, spelling, punctuation, and overall English of the manuscript. The scientific names of the species and the names of the genes must be italicized in the manuscript. The abbreviations should be fully explained during the first mention in the abstract and introduction. Before recommending this article for publication, there are some shortcomings that should be resolved.

General comments

Overall, the study is well designed and presented in a good way, but mostly the literature is not cited.  

Abstract

Add one or two lines of conclusion and future perspective of the study.

The methodology should be brief and clear.

Also, add quantitative results in this section.

Change 3,205.63 km2, into Km2

Introduction

The introduction part is well written, but still some details are required. The authors should provide details of the distribution of the studied species in the world as well as in Mexico.

Threats to the conservation of this species in a few sentences.

What is the link of this species with SARS-CoV-2?

After the first use of full name, use the name like C. aequipetala Cav. Throughout the manuscript.

Provide economic and commercial importance of the C. aequipetala Cav.

Line 43 and 45 must be cited.

The following sentence should be cited with relevant literature doi: 10.1002/ptr.6787 “The pandemic caused by the SARS-CoV-2 virus has abruptly reminded us of the importance of the proper and responsible use of natural resources and, in general, the care of the planet”.

Line 90-91 should be cited with the following article. https://doi.org/10.1016/j.jep.2021.114515.

Add a few sentences on the significance of traditional uses and phytochemicals of the plants. The following article may be cited. https://doi.org/10.1016/j.chnaes.2021.03.009. 

Materials and methods

Methodology is well written. Line 146 to 150 should be revised and clarified.

Results

Line 180-183 must be clear and readable.

Discussion

Compare the obtained results with the more current study.

Conclusion

Conclusion is well justified.  

TRANSLATE with x English
Arabic Hebrew Polish
Bulgarian Hindi Portuguese
Catalan Hmong Daw Romanian
Chinese Simplified Hungarian Russian
Chinese Traditional Indonesian Slovak
Czech Italian Slovenian
Danish Japanese Spanish
Dutch Klingon Swedish
English Korean Thai
Estonian Latvian Turkish
Finnish Lithuanian Ukrainian
French Malay Urdu
German Maltese Vietnamese
Greek Norwegian Welsh
Haitian Creole Persian  
TRANSLATE with COPY THE URL BELOW Back EMBED THE SNIPPET BELOW IN YOUR SITE Enable collaborative features and customize widget: Bing Webmaster Portal Back

Reviewer 2 Report

Overall I find your paper interesting but requiring revision.

Please find below some of my comments and suggestions.

Lines 51 and 52: add the local names in Italics of cancer weed, blow weed, or weed that arises from the water...., you can keep the English translations, after each vernacular name, between parentheses. The same apply for lines 224 to 226.

As a rule use in all the Figure and Table captions Cuphea instead of C. in order to make these more auto explicative

Represent in one map the eight localities mentioned in Table 2 where your ethnobotanical information was recorded. Perhaps you can use the map of Figure 2 to represent not only potential distribution of the species but also the distribution of the knowledge recorded

I suggest to replace some words or expressions especially in tables

Table 2

Older adults   replace with    Elders

Table 3.

Medicinal uses of C. aequipetala      replace with        Medicinal uses of Cuphea aequipetala

Whas wounds replace with Wound washing

Cicatrization   replace with    To promote wound healing

Hits    replace with    Bumps and bruises

Pain (bones, teeth, head and throat)   replace with   Headache, tooth, bone or throat pain

Cancer  replace with Tumors

Stomach disordes      replace with      Stomach upsets and disorders

Twisting       replace with        Sprain

Figure 6.

Please add in the caption the explanation of the abbreviations used in the figure for the different climate types

Discussion

Given the detailed study of potential distribution of the species I miss in the discussion an assessment of the localities reported in the GBIF.org map for the species that seem exceed largely the potential area. Is this due to cultivation? Furthermore, the “Mapa de areas de uso” in the monograph of the Biblioteca Digital de la Medicina Tradicional Mexicana considering you map of potential areas and the localities where you conducted ethnobotanical research.

Reviewer 3 Report

Dear editor

Ecological modeling as well ethno botany are the key subjects in conservation management. Regardless there are several shortcoming in introduction, methodology, results and discussion.

Title

Why are two different and somewhat unrelated approaches to biodiversity management presented together?

Abstract

This section do not cover all parts of study. After the revise authors should be improve it based on corrected manuscript. Please consider key achievement, justification and main aims in abstract

Introduction

What is relation of this sentences to biodiversity loss? I think the hypotheses are not represented clearly

There is a non-connection between last and next sentences. The authors should be consider it

Authors should be design a better introduction including: hypothesis, necessity, aims and etc.

Introduction does not justify the importance of study

Literature review showing several shortcomings

In this section, writers should pay attention to following  points:

Some descriptions on distribution patterns as well ecological modeling as well its importance to biodiversity management.

Stronger justification for the importance of herbs and medicinal plants

Improve the literature review

Study area

Study are showing following shortcomings:

The high resolution maps of study area

Detailed description of the ecological features of the region and its place in the world  Geology, geomorphology , geography, climatology and ecology of study area

Why these methods is used? The type of data? Limitations and etc.?

What was the basis of your sampling in this study?

Have you used raster layers to do modeling? How much is the resolution of these layers? Please add the necessary comments to the text.

How were these factors (ecological modeling) selected? Field work experiments, ...?

 Describe the method exactly

In ethnobotanical studies, details related to plant collection are also important. Please add information about plant samples to Table 2 or submit them to the journal in the supplementary file. Information such as the voucher number, locality, as well as the herbarium where the specimens are kept should be added to the manuscript.

Please add the AUC (area under ROC) curves in developing habitat suitability model to the text.

Please add the image of the results of jackknife evaluations of the relative importance of the predictor variables and their percentage contribution.

You mentioned in the method that you used 22 variables. Which one is correct 23 or 22

Please write the code for the variables in Table1 and the text in a form. All in the form of BIO or Bio. Please apply this throughout the text.

Please write the code for the variables in Table1 and the text in a form. All in the form of BIO or Bio. Please apply this throughout the text.

Please edit the legend and colors on the map (fig 4). The colors in the legend do not match the map.

Please edit the legend and colors on the map(fig 5). The colors in the legend do not match the map

Please edit the legend and colors on the map. The colors in the legend do not match the map.

 Discussion

Please add the author name of the species to the scientific name

You mentioned in the method that you used 22 variables. Which one is correct 23 or 22?

What is the relationship between the results of ethnobotany and ecological modeling in this study? You have gathered medicinal information about this plant during an ethnobotanical survey. How can this help prevent the extinction of this species and be a solution to protect it?

Discussion should be represent the main achievements clearly, so this goal is achieved in the form of comparisons with previous studies and critiques of achievements

Conclusion

What solutions do you suggest for such protection? Please add to this section.

The manuscript needs to be revised in terms of editing. Check the grammar thoroughly. Upgrade text.

Finally, The manuscript is not acceptable in its current form and should be reviewed after careful review and structural modifications

Author Response

Dear editor

Ecological modeling as well ethno botany are the key subjects in conservation management. Regardless there are several shortcoming in introduction, methodology, results and discussion.

Title Why are two different and somewhat unrelated approaches to biodiversity management presented together?

R. They are presented together because are important elements in the management and conservation of Biocultural diversity.

Abstract This section do not cover all parts of study. After the revise authors should be improve it based on corrected manuscript. Please consider key achievement, justification and main aims in abstract

R. This has been done as suggested by the reviewer and the abstract has been rewritten.

Introduction What is relation of this sentences to biodiversity loss?

R. In order not to have information that does not contribute to the manuscript, we eliminate this sentence from the document.

I think the hypotheses are not represented clearly

R. The hypothesis has been rewritten and is now clear.

There is a non-connection between last and next sentences. The authors should be consider it

R. The introduction section has been fully rewritten and the information between sentences is now connected.

Authors should be design a better introduction including: hypothesis, necessity, aims and etc.

R. The introduction section has been fully rewritten, and the information provided is now clear.

Introduction does not justify the importance of study

R. The introduction section has been fully rewritten, and the information provided is now clear.

Literature review showing several shortcomings

In this section, writers should pay attention to following points: Some descriptions on distribution patterns as well ecological modeling as well its importance to biodiversity management.

R. The introduction section has been fully rewritten, and the information provided is now clear.

Stronger justification for the importance of herbs and medicinal plants Improve the literature review

R. The introduction section has been fully rewritten, and the information provided is now clear.

Study area

Study area showing following shortcomings: The high resolution maps of study area

R. This has been done as suggested by the reviewer, the map resolution is now the right one.

Why these methods is used? The type of data? Limitations and etc.?
R. This information is included in the introduction section and M&M section.

What was the basis of your sampling in this study?
R. No plant sampling was performed for this study, the 90 in situ presence data (geobraphic coordinates) were obtained with a field GPS navigator; brand Garmin model etrex22 < 3 m precision. This information is described in the M&M section.

Have you used raster layers to do modeling? How much is the resolution of these layers? Please add the necessary comments to the text.

R. Raster layers were not used to model. Shape layers (.shp) were used. The necessary information is described M&M section.

How were these factors (ecological modeling) selected?
R. This information is now included in the M&M section.

Field work experiments, ...? Describe the method exactly
R. The field work performed in this work consisted of obtaining geographic coordinates, photographs of populations and plants for identification, in addition to the application of surveys. This information is described in the M&M section.

In ethnobotanical studies, details related to plant collection are also important. Please add information about plant samples to Table 2 or submit them to the journal in the supplementary file.

R. No plant sampling was performed for this study.

Information such as the voucher number, locality, as well as the herbarium where the specimens are kept should be added to the manuscript.
R. This information is now included in the M&M section.

Please add the AUC (area under ROC) curves in developing habitat suitability model to the text.

R. This has been done as suggested by the reviewer (Figure 3).

Please add the image of the results of jackknife evaluations of the relative importance of the predictor variables and their percentage contribution.

R. The jackknife evaluation figure was added as supplementary material (Figure S1).

You mentioned in the method that you used 22 variables. Which one is correct 23 or 22

R. We used 22 variables in the study. The manuscript was carefully reviewed, and this error was corrected.

Please write the code for the variables in Table1 and the text in a form. All in the form of BIO or Bio. Please apply this throughout the text.

R. This has been done as suggested by the reviewer

Please edit the legend and colors on the map (fig 4). The colors in the legend do not match the map. Please edit the legend and colors on the map(fig 5). The colors in the legend do not match the map Please edit the legend and colors on the map. The colors in the legend do not match the map.

R. This has been done as suggested by the reviewer

Discussion

Please add the author name of the species to the scientific name

R. This has been done as suggested by the reviewer

You mentioned in the method that you used 22 variables. Which one is correct 23 or 22?
R. We used 22 variables in the study. The manuscript was carefully reviewed, and this error was corrected.

What is the relationship between the results of ethnobotany and ecological modeling in this study?
R. This analysis was carried out by comparing the potential distribution of C. aequipetala obtained in this study with the maps, both of the localities reported for the species in the GBIF.org, and with the "Map of areas of use" of the medicinal plant C. aequipetala in the monograph of the Digital Library of Traditional Mexican Medicine.

You have gathered medicinal information about this plant during an ethnobotanical survey. How can this help prevent the extinction of this species and be a solution to protect it?

R. This has been done as suggested by the reviewer.

Discussion should be represent the main achievements clearly, so this goal is achieved in the form of comparisons with previous studies and critiques of achievements.

R. This has been done as suggested by the reviewer.

Conclusion

What solutions do you suggest for such protection? Please add to this section.

R. This has been done as suggested by the reviewer.

The manuscript needs to be revised in terms of editing. Check the grammar thoroughly. Upgrade text.
R. The manuscript was revised by the English editing MDPI services.

Round 2

Reviewer 3 Report

Dear editor

All comments have been done

Best regards